# Designing Mechanical Meta-Materials by Learning Equivariant Flows

**Mehran Mirramezani**
Department of Computer Science
Princeton University
m.mirramezani@berkeley.edu

**Anne S. Meeussen & Katia Bertoldi**
School of Engineering and Applied Sciences
Harvard University
{meeussen,bertoldi}@seas.harvard.edu

**Peter Orbanz**
Gatsby Computational Neuroscience Unit
University College London
p.orbanz@ucl.ac.uk

**Ryan P. Adams**
Department of Computer Science
Princeton University
rpa@princeton.edu

## Abstract

Mechanical meta-materials are porous solids whose geometric structure results in exotic nonlinear mechanical behaviors that are not typically achievable via homogeneous materials. We show how to drastically expand the design space of a class of mechanical meta-materials known as *cellular solids*, by generalizing beyond translational symmetry of a unit pore cell. This is made possible by transforming a reference geometry according to a divergence free flow that is parameterized by a neural network and equivariant under the relevant symmetry group. We show how to construct flows equivariant to the space groups, despite the fact that these groups are not compact. Coupling this flow with a differentiable nonlinear mechanics simulator allows us to represent a much richer set of cellular solids than was previously possible. These materials can be optimized to exhibit desirable mechanical properties such as negative Poisson's ratios or to match target stress-strain curves. We validate simulated mechanical behaviors of these new designs against fabricated real-world prototypes. We find that designs with higher-order symmetries can exhibit a wider range of behaviors.

## 1 Introduction

Mechanical meta-materials are engineered *porous* structures in which geometric features of the pores lead to uncommon mechanical behaviors not achievable from natural materials (Bertoldi et al., 2017) with various applications in soft robotics (Khajehtourian & Kochmann, 2021), biomedical devices (Wang et al., 2023), etc. One promising class of mechanical meta-materials are *cellular solids*, characterized by an array of pores, that exhibit exotic nonlinear properties such as auxeticity (expanding under tension) (Bertoldi et al., 2010) and reversible shape morphing (Wenz et al., 2021). The nonlinear functionalities of cellular solids can be programmed by: i) optimizing the geometry of the pores, and/or ii) manipulating the arrangement of the pores.

Existing approaches to cellular solids design have been focused on the former, optimizing the shape (Overvelde & Bertoldi, 2014; Xue & Mao, 2022; Medina et al., 2023) or topology (Wang et al., 2014; Gao et al., 2019) of the pores of a cellular solid constructed by *translational* symmetric arrangement of a single pore unit cell. The design of such cellular solids has also begun to attract substantial attention from the machine learning community, e.g., Beatson et al. (2020); Ma et al. (2020); Kollmann et al. (2020); Ha et al. (2023). Figure 1(a) shows examples of such cellular solids designs from Overvelde et al. (2012). However, the space of potential symmetric arrangement of the pores is not limited only to translation, and exploiting other spatial patterns of pores can potentially unlock new cellular solids with unexpected mechanical responses. Representing and optimizing this larger space of cellular solids is challenging, but recent developments in machine learning of crystallographically-invariant functions (Adams & Orbanz, 2023) make it possible to construct unconstrained parameterizations of the symmetric arrangement of the pores that can be used for

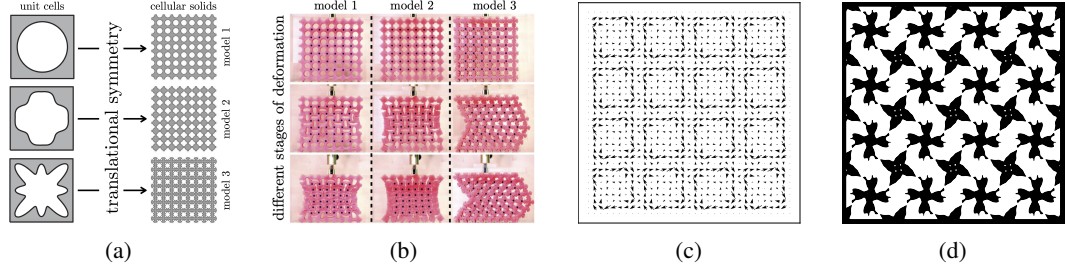

Figure 1: (a) Example of three shape functions $s : \Omega \subset \mathbb{R}^2 \to \{0, 1\}$ constructed by translational symmetric arrangement of different unit pore cells, from Overvelde et al. (2012). (b) Real-world experiment demonstrating different nonlinear behaviors of fabricated cellular solids specified by the shape functions in (a), from Overvelde et al. (2012). (c) A vector field defined by a flow equivariant under the space group p4, constructed as in Section 3. (d) A design based on a level set of the p4-invariant function in (c). Note that it is disconnected.

shape optimization. In this work, we leverage these models and introduce a novel framework for learning richer classes of cellular solids, in which we not only optimize pore shapes but also explore all possible arrangements of the pores from two-dimensional crystallographic symmetry groups.

The approach we take to the problem is to define the geometry *implicitly* via a neural network flow from a reference configuration. By carefully constructing this flow to preserve group invariance, that enables all 17 crystallographically symmetric arrangement of the pores including translation, simple symmetric cellular structures can be transformed parametrically into complex cellular structures, as illustrated in Figure 3. Another advantage of this approach is that it ensures the solution avoids disconnected regions and associated numerical challenges observed in existing techniques such as level set method (Wang et al., 2014) shown in Figure 1(d) or moving morphable voids method (Du et al., 2024). We additionally show that such neural flows can be made divergence-free, which makes it possible to avoid modifying the total volume of the structure, allowing the optimization to focus purely on geometry as the means of solving the task.

The paper is structured as follows. We first introduce cellular solids, framed in terms of a design space that can exhibit symmetries. Section 3 then develops a constrained optimization problem for learning shapes which are symmetric with respect to a space group while achieving a target volume fraction. Our approach uses a novel formalism based on equivariant neural network flows. Next we provide an overview of our implementation, including details of the neural network, the differentiable mechanics solver, and the mechanical material model used. We then demonstrate the application of our proposed approach in designing cellular solids with nontrivial functionalities in both simulation and real experiments. We conclude with related work and a discussion on limitations and future steps.

## 2 CELLULAR SOLIDS AND SYMMETRIC SHAPES

An example of a cellular solid is illustrated in Figure 1(a). It consists of a homogeneous solid material, interrupted by empty regions that are called pores. Once the material is chosen, the solid is entirely characterized by its geometry. This geometry can be specified by a function $s : \Omega \to \{0, 1\}$, where $\Omega$ is the entire volume occupied by the solid, including pores. This function is called a **shape**. A value $s(x) = 1$ specifies there is material at location $x$, a value $s(x) = 0$ that $x$ is in a pore. We are particularly interested in cellular solids that are—like the example in Figure 1(a)—completely determined by a two-dimensional cross-section. We can then reduce the shape to a two-dimensional function $s : \Omega \subset \mathbb{R}^2 \to \{0, 1\}$, where $\Omega$ is now the area of the cross-section. The methods developed here can be used to construct shape functions on $\Omega \subset \mathbb{R}^n$ for both $n = 2$ and $n = 3$. Our simulations and real-world experiments assume $n = 2$.

**Periodic designs as tilings**. Clearly visible in Figure 1(a) is the periodic structure. This periodicity can be formalized as a tiling: If $\Pi \subset \mathbb{R}^n$ is a convex polytope, and $\mathbb{T}$ is a group of translations of $\mathbb{R}^n$, then $\mathbb{T}$ is said to **tile** the space with $\Pi$ if the images $\phi\Pi$ of the polytope under elements $\phi \in \mathbb{T}$ cover the entire space and only their boundaries overlap—formally, if

$$\bigcup_{\phi \in \mathbb{T}} \phi\Pi = \mathbb{R}^n \quad \text{and} \quad \phi\Pi \cap \psi\Pi = \text{empty set or face of } \phi\Pi \tag{1}$$

for all $\phi, \psi \in \mathbb{T}$. If $\Pi$ is, for example, an axis-aligned square in $\mathbb{R}^2$, the group $\mathbb{T}$ would consist of all horizontal and vertical shifts by multiples of the edge length of $\Pi$. A shape $s$ is periodic if there is a function $\sigma : \Pi \to \{0, 1\}$ such that

$$s(x) = \sigma(\phi x) \qquad \text{if } x \in \phi\Pi \text{ for some } \phi \in \mathbb{T} . \tag{2}$$

In words, $\sigma$ specifies a pattern of pores on a small area $\Pi$, and this pattern is then replicated over $\Omega$ by shifts. To avoid dealing with the boundaries of $\Omega$ explicitly, we can define $s$ as a function $s : \mathbb{R}^2 \to \{0, 1\}$ on the entire plane, and restrict it to $\Omega$ where necessary (see Section 4 for details); observe that (2) indeed specifies $s$ on all of $\mathbb{R}^2$, if $\mathbb{T}$ tiles $\mathbb{R}^2$ with $\Pi$. We also note that $s$ satisfies (2) if and only if it is invariant under $\mathbb{T}$,

$$s(\phi x) = s(x) \qquad \text{for all } \phi \in \mathbb{T} \text{ and } x \in \mathbb{R}^2 ,$$

in which case $\sigma$ is the restriction of $s$ to $\Pi$.

**Expanding the design space**. Our point of departure from existing designs is the observation that tilings cannot only be generated by translations. In general, one may substitute $\mathbb{T}$ by a group $\mathbb{G}$ of isometries of $\mathbb{R}^n$. Such a group is said to tile $\mathbb{R}^n$ with a convex polytope $\Pi$ if (1) holds with $\mathbb{G}$ substituted for $\mathbb{T}$. Any group of isometries that tiles $\mathbb{R}^n$ with a convex polytope $\Pi$ is called a **space group** or **crystallographic group** on $\mathbb{R}^n$. We are particularly interested in the case $n = 2$ that are also known as **wallpaper groups**. In addition to translations, space groups may also contain rotations, reflections, and other isometries. Our approach is to construct cellular solids with shape functions that satisfy the generalized periodicity

$$s(\phi x) = s(x) \qquad \text{for all } \phi \in \mathbb{G} \text{ and } x \in \mathbb{R}^2 , \tag{3}$$

for a space group $\mathbb{G}$. We call such a shape **symmetric** with respect to $\mathbb{G}$.

**Properties of space groups**. The two-dimensional space groups are completely classified: Two space groups $\mathbb{G}_1$ and $\mathbb{G}_2$ are **isomorphic** if one can be transformed into the other by an affine deformation of the underlying space, i.e., if there is an affine, order-preserving bijection $\alpha : \mathbb{R}^n \to \mathbb{R}^n$ such that $\phi \in \mathbb{G}_1$ if and only if $\alpha\phi\alpha^{-1} \in \mathbb{G}_2$. Up to isomorphism, there are only finitely many space groups for each dimension $n$. For $n = 2$, there are 17 such groups. These are illustrated in Appendix A.1 Some properties of space groups relevant in the following are:

(i) Each space group $\mathbb{G}$ is countably infinite.

(ii) Since each element $\phi$ of $\mathbb{G}$ is an isometry, it is of the form $\phi x = A_\phi x + b_\phi$, where $A_\phi$ is an orthogonal matrix and $b_\phi \in \mathbb{R}^n$.

(iii) $\mathbb{G}$ contains a countably infinite subgroup of pure translations, namely

$$\mathbb{T}_\mathbb{G} := \{\phi \in \mathbb{G} \,|\, A_\phi = \text{identity}\} = \{\phi \in \mathbb{G} \,|\, \phi(x) = x + b_\phi \text{ for some } b_\phi \in \mathbb{R}^n\} .$$

This group is always infinite, and generated by $n$ linearly independent shifts: There exist $n$ linearly independent vectors $b_1, \ldots, b_n \in \mathbb{R}^n$ such that

$$\phi \in \mathbb{T}_\mathbb{G} \qquad \Longleftrightarrow \qquad \phi(x) = x + \sum_{i=1}^{n} c_i(\phi)b_i \quad \text{for some} \quad c_1(\phi), \ldots, c_n(\phi) \in \mathbb{Z} .$$

We refer to Vinberg & Shvartsman (1993); Ratcliffe (2006) for more on the geometry of such groups, and Hahn et al. (1983) for their crystallographic properties.

## 3 SYMMETRIC SHAPES FROM SYMMETRY-PRESERVING FLOWS

We select a class of symmetries by fixing a space group $\mathbb{G}$. Our goal is to solve two problems:

    I. Constructively define a set $\mathcal{S} = \{s_\theta \,|\, \theta \in \Theta\}$ of shapes that satisfy (3), indexed by parameters in some parameter space $\Theta$.

    II. Given a loss function $\ell : \text{shape} \to \mathbb{R}$, solve $\theta^* := \arg\min_{\theta \in \Theta} \ell(s_\theta)$.

The loss represents how well the shape achieves some physical property, such as the Poisson's ratio illustrated in Figure 5. To solve (I), we proceed as follows:

- We start with a symmetric reference shape $s_0$. This is a $\mathbb{G}$-invariant function $s_0 : \mathbb{R}^n \to \{0, 1\}$ that satisfies (3).

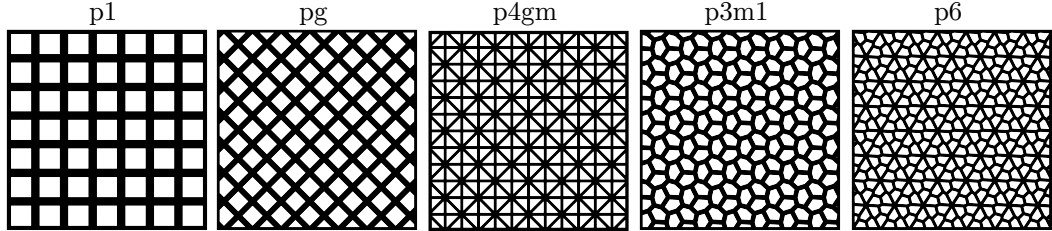

Figure 2: Examples of the reference shape $s_0$ for different space groups. The labels `p1`, `pg`, etc. refer to the crystallographic naming standard for such groups see Appendix A.1.

- We then construct a class $\{F_\theta | \theta \in \Theta\}$ of flows that are symmetry- and volume-preserving.
- Each shape $s_\theta$ is defined as the deformation of $s_0$ by $F_\theta$. That is, we fix a "terminal time" $t_{\max} > 0$, and define

$$s_\theta(x) := s_0(F_\theta(x, t_{\max})) .$$

These functions are shapes on $\mathbb{R}^n$ that satisfy (3), and can be restricted to shapes on $\Omega$.

The remainder of this section explains this construction in detail. This solution of (I) can then be combined with a standard differentiable mechanics simulator to solve (II), as described in Section 4.

### 3.1 SYMMETRY- AND VOLUME-PRESERVING FLOWS

A flow is a function $F : \mathbb{R}^n \times \mathbb{R}_+ \to \mathbb{R}^n$ that satisfies

$$F(x, 0) = x \quad \text{and} \quad F(x, s + t) = F(F(x, s), t) \qquad \text{for } x \in \mathbb{R}_n \text{ and } s, t \in \mathbb{R}_+ .$$

For our purposes, we can restrict the second argument of $F$ to an interval $I = [0, t_{\max}]$. We call a flow **symmetry-preserving** for a space group $\mathbb{G}$ if it is $\mathbb{G}$-equivariant pointwise in time,

$$\phi \circ F(x, t) = F(\phi x, t) \qquad \text{for all } \phi \in \mathbb{G} \text{ and } (x, t) \in \mathbb{R}^n \times I .$$

If that is the case, then $s_\theta$ is $\mathbb{G}$-invariant, since $s_0$ is $\mathbb{G}$-invariant and so

$$s_\theta(\phi x) = s_0(F_\theta(\phi x, t_{\max})) = s_0(\phi F_\theta(x, t_{\max})) = s_0(F_\theta(x, t_{\max})) = s_\theta(x) .$$

We also require $F$ to be **volume-preserving**. A continuous function $g : \mathbb{R}^n \to \mathbb{R}^n$ is volume-preserving if $\mathrm{vol}(g(R)) = \mathrm{vol}(R)$ for every (measurable) set $R \subset \mathbb{R}^n$. For a flow $F$, this means

$$\mathrm{vol}(F(R, t)) = \mathrm{vol}(R) \qquad \text{for every (measurable) set } R \subset \mathbb{R}^n \text{ and } t \in I .$$

This ensures that the ratio of volume occupied by pores does not change if a shape is deformed using $F$.

It is well known that a flow $F$ can be constructed as the solution of a differential equation. If $H : \mathbb{R}^n \times I \to \mathbb{R}^n$ is a (sufficiently smooth) function, the constrained equation

$$\frac{d}{dt} F(x_0, t) = H(F(x_0, t), t) \quad \text{subject to } F(x_0, 0) = x_0 \qquad \text{for } t \in I, x_0 \in \mathbb{R}^n \qquad (4)$$

has a unique solution $F$, and this solution is a flow. Each smooth function $H$ hence determines a flow $F$, and we can specify a parameterized class of flows $F_\theta$ by specifying a parameterized class of smooth functions $H_\theta$. Theorem 2 below shows that the flow $F$ is volume- and symmetry-preserving if $H$ is volume-preserving and satisfies

$$H(\phi x, t) = A_\phi H(x, t) \qquad \text{for } \phi \in \mathbb{G}, x \in \mathbb{R}^n, t \in I . \qquad (5)$$

To construct such a function $H$, we define a symmetrization operator $\Gamma$ and an operator $\Lambda$ that turns functions into volume-preserving functions.

### 3.2 VOLUME PRESERVATION

To define $\Lambda$, we can draw on existing work: A continuously differentiable function $g$ is volume-preserving if and only if it is divergence-free,

$$(\mathrm{div} \, g)(x) = (\nabla \cdot g)(x) = 0 \qquad \text{for all } x \in \mathbb{R}^n .$$

It was noted in Richter-Powell et al. (2022) that, if $g : \mathbb{R}^n \to \mathbb{R}^{n(n-1)/2}$ is a function that is twice continuously differentiable then

$$(\Lambda g)(\mathbf{x}) := \begin{bmatrix} 0 & \partial_2 g_1(\mathbf{x}) & \partial_3 g_2(\mathbf{x}) & \cdots & \partial_n g_{n-1}(\mathbf{x}) \\ -\partial_1 g_1(\mathbf{x}) & 0 & \partial_3 g_n(\mathbf{x}) & \cdots & \partial_n g_{2n-3}(\mathbf{x}) \\ -\partial_1 g_2(\mathbf{x}) & -\partial_2 g_n(\mathbf{x}) & 0 & \cdots & \partial_n g_{3n-6}(\mathbf{x}) \\ \vdots & \vdots & \vdots & \ddots & \vdots \\ -\partial_1 g_{n-1}(\mathbf{x}) & -\partial_2 g_{2n-3}(\mathbf{x}) & -\partial_3 g_{3n-6}(\mathbf{x}) & \cdots & 0 \end{bmatrix} \cdot \begin{bmatrix} 1 \\ 1 \\ 1 \\ \vdots \\ 1 \end{bmatrix} \tag{6}$$

is a divergence-free function $\Lambda g : \mathbb{R}^n \to \mathbb{R}^n$.

### 3.3 SYMMETRIZATION

For finite groups, invariant or equivariant functions are typically constructed by the "summation trick", which starts with a function $f$ and sums over all compositions $f \circ \phi$ with group elements. This is not possible for space groups, since these groups are countably infinite. We can, however, decompose the symmetrization into two steps. This decomposition is based on the following property.

**Theorem 1.** *Let $\mathbb{G}$ be a space group on $\mathbb{R}^n$. For each $\phi \in \mathbb{G}$, denote by $c_i(\phi)$ the linear coefficient of $b_\phi$ with respect to the shift basis vector $b_i$, that is, $\phi(x) = A_\phi x + \sum_{i \le n} c_i(\phi) b_i$. Then*

$$\widehat{\mathbb{G}} := \{\phi \in \mathbb{G} \mid c_1(\phi), \dots, c_n(\phi) \in [0,1)\}$$

*is a finite subset of $\mathbb{G}$. For each $\phi \in \mathbb{G}$, there are unique elements $\hat\phi \in \widehat{\mathbb{G}}$ and $\tau_\phi \in \mathbb{T}_\mathbb{G}$ such that $\phi = \hat\phi + \tau_\phi$. If $f : \mathbb{R}^n \to \mathbb{R}^n$ is invariant under the shift subgroup $\mathbb{T}_\mathbb{G}$ of $\mathbb{G}$, the function*

$$(\Gamma f)(\mathbf{x}) := \frac{1}{|\widehat{\mathbb{G}}|} \sum_{\phi \in \widehat{\mathbb{G}}} \mathbf{A}_\phi^{-1} f(\phi x)$$

*satisfies $A_\phi(\Gamma f) = (\Gamma f) \circ \phi$ for each $\phi \in \mathbb{G}$.*

*Proof.* See Appendix A.2.1. $\square$

To construct the $\mathbb{T}_\mathbb{G}$-invariant function required by the result, we use maximal invariants: Define an $n \times n$-matrix $B$ as

$$B := \text{ matrix whose columns are the lattice basis vectors } b_1, \dots, b_n.$$

This matrix describes a change of basis from the cartesian basis to $b_1, \dots, b_n$. The set $\{B^{-1}\phi \mid \phi \in \mathbb{T}_\mathbb{G}\}$ therefore consists of all shifts by vectors with integer entries, or in other words, $B^{-1}\mathbb{T}_\mathbb{G} = \mathbb{Z}^n$. Define a function $\rho : \mathbb{R}^n \to \mathbb{R}^{2n}$ as

$$\rho(x) := \tilde\rho(B^{-1}x) \quad \text{where} \quad \tilde\rho(u) := (\cos(2\pi u_1), \sin(2\pi u_1), \dots, \cos(2\pi u_n), \sin(2\pi u_n))^\mathsf{T}.$$

This function maps $\mathbb{R}^n$ onto the $2n$-dimensional torus. It is well known that $\tilde\rho$ is a maximal invariant for the shift group $\mathbb{Z}^n$. This means that a function $g$ on $\mathbb{R}^n$ is $\mathbb{Z}^n$-invariant if and only if it can be represented as $g = g' \circ \rho$ for some function $g'$ on $\mathbb{R}^{2n}$. It follows that $\rho$ is a maximal invariant for $\mathbb{T}_\mathbb{G}$.

### 3.4 FLOW CONSTRUCTION

We can now combine $\Gamma$, $\Lambda$ and $\rho$ to construct a divergence-free function $H$ that satisfies (5) from a smooth function $h$, as the next result shows. We can therefore use a neural network with parameter vector $\theta$ to represent a class $\{h_\theta | \theta \in \Theta\}$ of such functions. For each $\theta$, we then obtain a unique flow $F_\theta$ that preserves volume and symmetry.

**Theorem 2.** *Let $\mathbb{G}$ be a space group on $\mathbb{R}^n$. Choose a function $h : \mathbb{R}^{2n} \times I \to \mathbb{R}^{n(n-1)/2}$ that is Lipschitz-continuous, and $(k+1)$-times continuously differentiable on $I$, for some $k \in \mathbb{N}$. Then*

$$H(x, t) := \Gamma(\Lambda(h(\rho(x), t))) \qquad \text{for } (x,t) \in \mathbb{R}^n \times I$$

*is a divergence-free function $\mathbb{R}^n \times I \to \mathbb{R}^n$ that satisfies (5). For this function $H$, the differential equation (4) has a unique solution $F$. The function $F$ is a flow, is volume-preserving, and is symmetry-preserving for $\mathbb{G}$. It is continuous in its first argument, and $k$ times continuously differentiable in the second.*

*Proof.* See Appendix A.2.2. $\square$

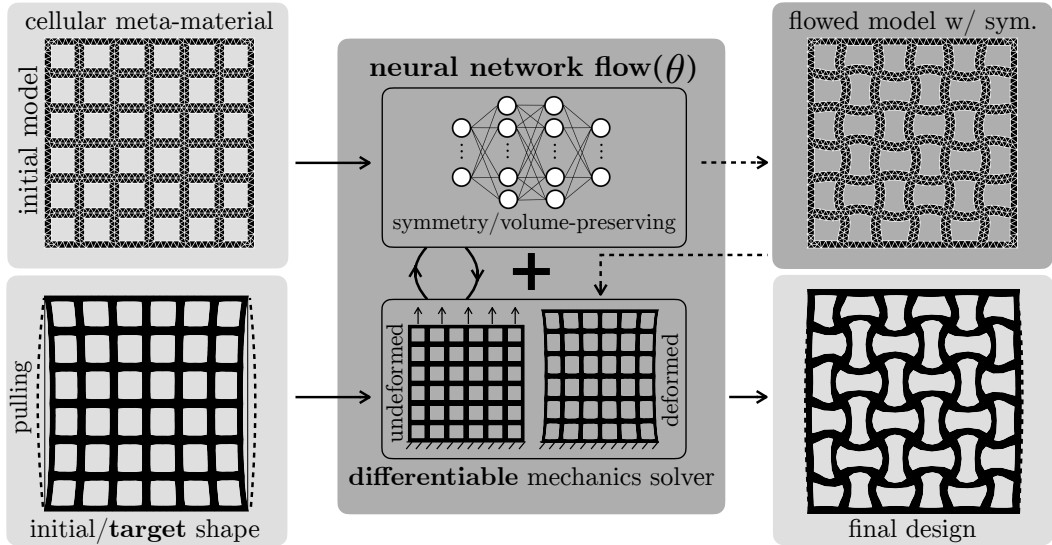

Figure 3: An overview of our modeling framework implemented in JAX. An equivariant neural network parameterized by $\theta$ works in tandem with a differentiable mechanics solver to flow an initial geometry to learn cellular solids with desired functionalities from a rich set of volume- and symmetry-preserving geometries. The upper left shape is $s_0$ and the upper right shape is $s_\theta$.

## 4 IMPLEMENTATION

To design cellular solids, we combine the shape representation defined above with a mechanics simulator. We proceed as follows:

1) Start with a reference shape $s_0$. This is generated by selecting a space group $\mathbb{G}$ and a tiling convex polytope $\Pi$ for this group. We then define a set of pores on $\Pi$ and tile with the images $\phi\Pi$ under group elements $\phi \in \mathbb{G}$ such that the entire region $\Omega$ is covered.

2) The region $\{x : s_0(x) = 1\}$ is represented by a triangular mesh using pygmsh, an open-source python package for geometry and mesh processing. This yields a discretization of the shape in terms of $N$ points in $\mathbb{R}^2$.

3) The image of the mesh points under a flow $F_\theta$ is constructed as in Section 3 to define a symmetric shape covering the same volume as $s_0$ for each parameter value $\theta$.

4) Using a mechanics simulator, we simulate the behavior of $s_\theta$ under physical effects, such as lateral deformation. We define a cost function $\ell$ that measures the response to a given physical effect, in such a way that small values of $\ell$ encode desired behavior.

5) We minimize $\ell$ with respect to $\theta$ using gradient descent, as described in more detail below.

The overall setup is illustrated in Figure 3.

**Differentiable mechanics simulator**. To solve the optimization problem, we must backpropagate gradients through both the mechanics solver and the flow representation. This involves solving a static nonlinear elasticity problem, described in Appendix A.3. To do so, we use an end-to-end differentiable continuum mechanics solver (Oktay, 2024). This solver is based on a variant of the finite element method called isogeometric analysis (IGA) (Hughes et al., 2005; Hughes, 2012). IGA represents both geometry and solutions in the same smooth B-spline basis functions. This allows us to directly construct differentiable maps from (i) geometry parameters to the solution, and (ii) to a specified loss function defined by the solution. The optimizer is implemented in JAX (Bradbury et al., 2018), and uses automatic differentiation and adjoint methods.

**Neural network ansatz**. The function $h_\theta$ in Section 3 is represented by a fully-connected neural network with two hidden layers of size 10, with tanh nonlinearity. To optimize these parameters, we use ADAM, with a learning rate of 0.001. Each optimization is performed several times with different initialization of the neural network parameters.

**Boundary conditions**. In Section 3, shapes are constructed as functions on $\mathbb{R}^n$, here for $n = 2$. To represent cellular solids, the function must be restricted to the finite volume $\Omega$, and we must ensure this restriction respects boundary conditions. To ensure preservation of the boundaries regardless of the parameters $\theta$, we use an envelope function which takes $H(x, t) \to 0$ at the edges of $\Omega$.

## 5 APPLICATIONS

Our experiments design novel cellular solids with two types of nonlinear behaviors: 1) nontrivial force-displacement responses, and 2) effective Poisson's ratios[1] under tension and compression. These types of behaviors are important in engineering applications such as soft robotics and programmable actuations. Here we explore examples of such behavior that would be challenging or impossible to achieve with homogeneous materials.

In uniaxial loading we apply a displacement uniformly to the top edge of the material while fixing the bottom edge; the left and right boundaries are free to move. The force-displacement response is usually described by a nominal stress-strain curve, in which nominal stresses ($S$) in our experiments are computed as the resulting vertical reaction forces per thickness (in the bottom boundary) normalized by the width of the cellular solids. Strain ($\epsilon$) values are the applied displacements per meta-material height ($H$) that in this study we use a displacement of equal to $10\%$ of the height of the structure resulting in the final applied strain of 0.1.

The effective Poisson's ratios $\nu_{ef}$ are approximated using horizontal displacement in a region of size $H/2$ in the middle of left and right boundaries ($\Omega_l$ and $\Omega_r$, see Figure 5 top left) as in Medina et al. (2023):

$$\nu_{ef} = \frac{1}{H\epsilon} \left( \frac{1}{|\Omega_l|} \int_{\Omega_l} u_x ds - \frac{1}{|\Omega_r|} \int_{\Omega_r} u_x ds \right). \tag{7}$$

When targeting force-displacement responses we use curves that are obtained at equally spaced displacement loading increments applied at the top edge to minimize a loss $L = \sum_i [S(\epsilon_i) - S^t(\epsilon_i)]^2$ with respect to a target $S^t$ response. We also design cellular solids with target Poisson's ratios at $\epsilon = 0.1$.

The mechanical behavior of cellular solids in this study is captured with a nearly incompressible neo-Hookean model with strain energy density function given in (13) that has material properties of Young's modulus $E = 50$ MPa and Poisson's ratio $\nu = 0.46$. In our experiments, cellular solids are of size $7 \times 7$ with initial material volume fraction of $50\%$ that is preserved during designs.

In all force-displacement experiments, neural network parameters are optimized until a loss value less than 0.0001 is achieved, where for Poisson's ratio designs a loss value less than 0.01 was used for stopping simulations. Using a single NVIDIA GeForce RTX 2080 Ti GPU, each optimization step (which requires forward simulation and gradient computation) requires approximately 60 seconds; there are at least $\sim$25,000 degrees of freedom in all mechanics models that found satisfactory through a mesh independence analysis described in Appendix A.4, such that simulation results were not affected by insufficient discretization resolution.

### 5.1 DESIGNING UNDER UNIAXIAL TENSION

Cellular solid structures built from hyperelastic materials usually exhibit nonlinear force-displacement responses. Here, we aim at proposing cellular solids with linear responses under uniaxially pulling experiments i.e., $S^t(\epsilon) = C\epsilon$ such that $S^t(\epsilon = 0.1) = \beta S_0$ where $\beta$ changes from 0.1 to 1.5 with an interval of 0.1 for a given normalized nominal stress $S_0$. We find that for all considered 17 symmetry groups, `p2gg` group indicated the best performance by being able to learn cellular solids with linear force-displacement responses for all $\beta$ values except $\beta = 1.5$. The learned meta-materials with their linear responses are depicted in Figure 4 for $\beta = 0.1$, 0.5 and 1.4. While we are able to achieve a wide range of linear responses with `p2gg` group, with `p1` translational symmetry we can design meta-materials that have mechanical responses with $\beta$ up to 1.1 (see Appendix A.5).

---

[1]Informally, Poisson's ratio is how much a material expands laterally when squeezed from the top, or contracts when pulled. Almost all materials have a positive Poisson's ratio.

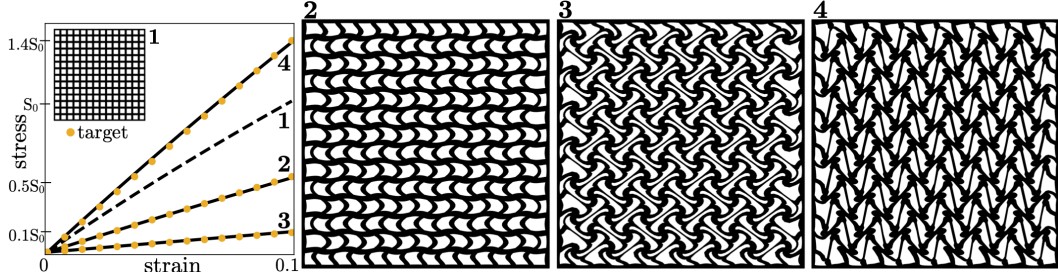

Figure 4: Undeformed configurations (i.e., $\epsilon = 0$) of three cellular solids designs with 50% volume fraction and pore shapes respecting `p2gg` symmetry group, and their linear stress-strain responses with the corresponding target curves during a uniaxial tension of the top edge up to $\epsilon = 0.1$.

Next we consider designing auxetic meta-materials with negative Poisson's ratios. We explored cellular solids that can exhibit a large $\nu_{ef} = -0.5$ during tension. With `p1` symmetry, the best design can only achieve $\nu_{ef} = -0.05$, whereas `pg` symmetry can improve this to $\nu_{ef} = -0.12$ (Figure 5). Our framework was able to learn a cellular solid with `p4` symmetry that has $\nu_{ef} = -0.45$ that is very close to the target negative Poisson's ratio. The eight best-performing cellular solids designs among the 17 group-constrained designs in achieving $\nu_{ef} = -0.5$ are shown in Table 1. This indicates the benefit of exploring a larger space of cellular solids than those exhibiting simple translational symmetries. In this case, six symmetry groups have better performances than `p1` symmetry. All final eight designs are depicted in Appendix A.6.

**Real-world experiments:** To demonstrate transfer to the real world, we manufactured the final designs with `pg` and `p4` symmetry groups and quantitatively verified their behavior under a uniaxial tension test (detailed in Appendix A.7). As predicted by our simulations in Figure 5 (second column) and confirmed with experimental observations (third column), the two proposed designs have negative Poisson's ratios. The experimental measurements $\nu_{ex} = -0.14$ and $\nu_{ex} = -0.49$ under pulling for `pg` and `p4` models, respectively, are in strong agreement with simulation results. The video footage of the `pg` cellular solid sample during pulling experiment with the corresponding simulation result are included in the supplementary materials.

## 5.2    DESIGNING UNDER UNIAXIAL COMPRESSION

The design of cellular solids under uniaxial compression is a more challenging task because of nonlinear buckling instabilities observed in such structures (Overvelde & Bertoldi, 2014). In such cases, the nominal stress-strain responses are characterized by a critical strain value ($\epsilon_{cr}$) at which mechanical responses vary. Here, we seek to design cellular solids that behave linearly before an $\epsilon_{cr}$ while having a constant nominal stress as the applied strain increases to the final value of 0.1:

$$S(\epsilon) = \begin{cases} C\epsilon & \text{if } \epsilon \leq \epsilon_{cr}, \\ C\epsilon_{cr} & \text{if } \epsilon_{cr} \leq \epsilon \leq 0.1. \end{cases} \tag{8}$$

Figure 6 illustrates mechanical responses of three cellular solid designs under uniaxial compression. We were able to design materials with various compression responses by controlling the plateau forces for a fixed $\epsilon_{cr}$ (designs 3 and 4) and tuning both the plateau force and the location of $\epsilon_{cr}$ (design 2) using `p1` symmetry, which found to be a challenging design factor in previous works (Medina et al., 2023). Interestingly, the only symmetry group that enabled such designs was `p1`. Examples of designs with other symmetries that force-displacement responses do not match target values are depicted in Appendix A.8.

Next, we focus on tailoring cellular solids with negative Poisson's ratios under compression. Here we aim to identify structures with $\nu_{ef} = -0.2$, i.e., requiring the material to shrink. To ensure this

Table 1: Poisson's ratios of cellular solids from eight symmetry groups with top performances when aiming for $\nu_{ef} = -0.5$.

| groups | p1 | p2 | pg | p2mg | p2gg | p4 | p3 | p6 |
|---|---|---|---|---|---|---|---|---|
| $\nu_{ef}$ | $-0.05$ | $-0.04$ | $-0.12$ | $-0.40$ | $-0.10$ | $-0.45$ | $-0.11$ | $-0.14$ |

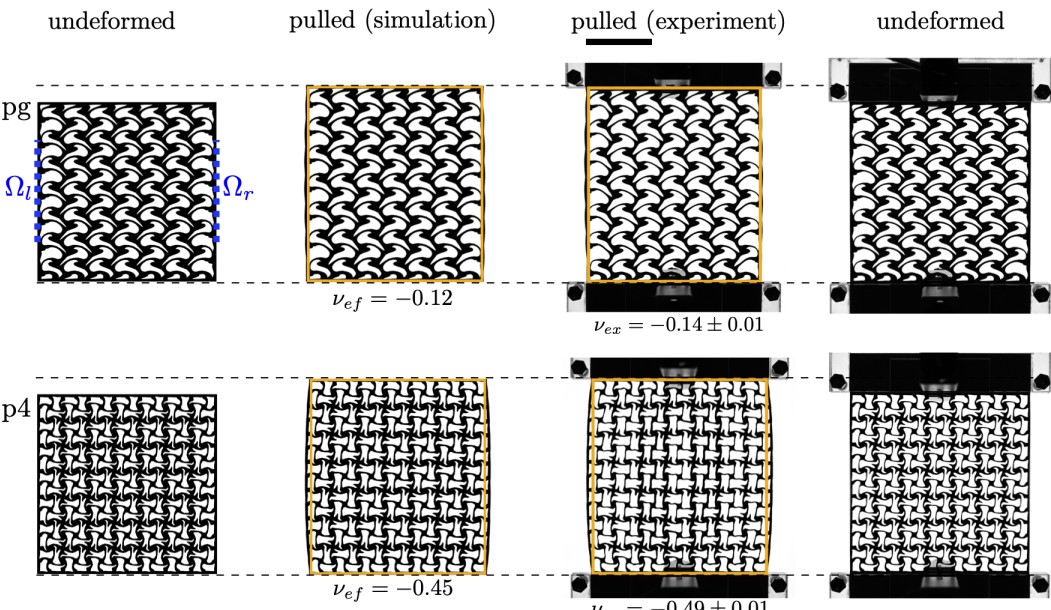

Figure 5: Undeformed configurations (i.e., $\epsilon = 0$) of two cellular solids designs that have negative Poisson's ratios with 50% volume fraction and pore shapes respecting pg and p4 symmetry groups (first column), and their deformed configurations from simulations during a uniaxial pulling of the top edge up to $\epsilon = 0.1$ (second column). Experimental realization of the same cellular solids under pulling confirms negative Poisson's ratios for both designs. The experimental measurements are also in strong numerical agreement with simulation results (third and fourth columns). Scale bars: 5cm.

design criterion, we customized the loss function to enforce points on $\Omega_l$ and $\Omega_r$ edges to have mean positive and negative displacements, respectively. We were able to optimize a p6 and pg symmetric cellular solids shown in Figure 7 with $\nu_{ef} = -0.15$ and $\nu_{ef} = -0.2$ under compression, respectively, that they also shrink.

Finally, we examine the ability of our framework in designing cellular meta-materials with a more complicated mechanical response: achieving zero effective Poisson's ratios under both uniaxial compression and tension via having zero displacements on $\Omega_l$ and $\Omega_r$ edges. Figure 8 shows the resulting successful structure that leverages p6 symmetry group, indicating zero displacement at both left and right edges for either pushing or pulling of 10% of the cellular solid height.

## 6 RELATED WORK

Existing cellular solids design are mainly focused on structures constructed from translational symmetry of an optimized pore unit cell. For example, Wang et al. (2014) utilized a level set method

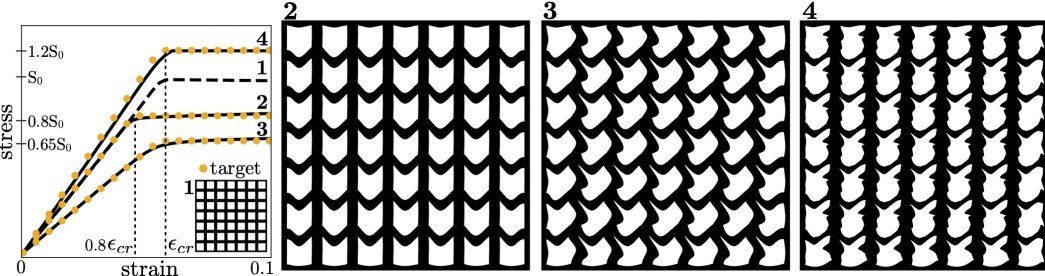

Figure 6: Undeformed configurations (i.e., $\epsilon = 0$) of three cellular solids designs with 50% volume fraction and pore shapes respecting p1 symmetry group, and their stress-strain responses with the corresponding target curves during a uniaxial compression of the top edge up to $\epsilon = 0.1$.

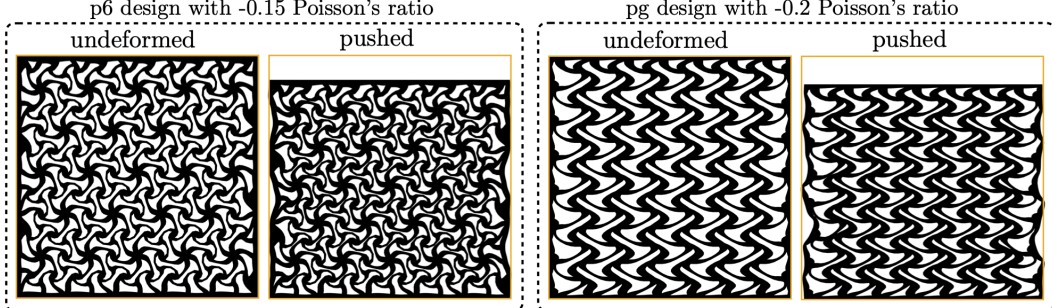

Figure 7: Undeformed and deformed configurations of a `p6` and `pg` symmetric cellular solids designs that have negative Poisson's ratios of $-0.15$ and $-0.2$, respectively, under uniaxial compression.

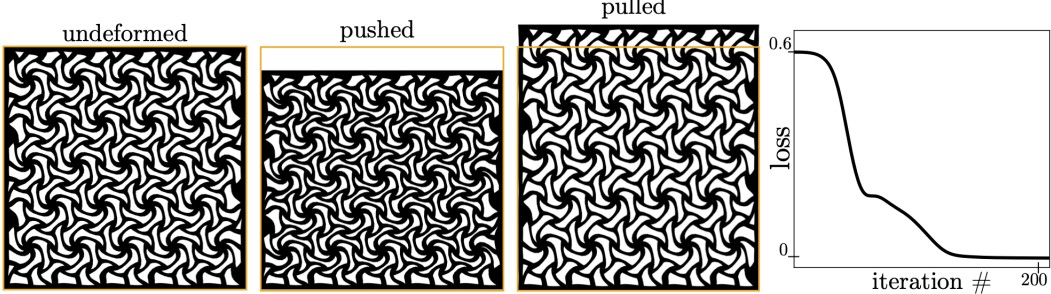

Figure 8: Undeformed and deformed configurations of a cellular solid design with 50% volume fraction and pore shapes respecting `p6` symmetry group indicating zero effective Poisson's ratios under both 10% uniaxial pushing and pulling, with loss values during optimization.

to design cellular meta-materials via unit cells topology optimization. Xue & Mao (2022) proposed an FEM-based mapped shape optimization technique to design periodic meta-materials. Medina et al. (2023) employed a higher order moving-mesh method to optimize the pore shapes of cellular solids. Du et al. (2024) optimized pore structures through describing them by a moving morphable voids method. Machine learning methods are becoming an emerging tool for both accelerating the design process and discovering new meta-materials beyond intuitions. For example, Xue et al. (2020b) introduced a data-driven neural network homogenization approach for cellular meta-materials design. Ma et al. (2020) applied machine learning tools for characterization of non-uniform arrangement of cells in meta-materials. Ha et al. (2023) leveraged generative machine learning techniques for inverse design of meta-materials with target stress-strain responses. In this study, we propose an equivariant neural network flow to learn volume-preserving cellular solids via performing shape optimization of the pores while exploring all 2D crystallographic symmetry groups to find the optimal arrangement of the pores.

## 7 LIMITATIONS AND FUTURE WORK

We introduce a machine learning framework that enables an efficient approach for inverse design of two-dimensional cellular solids with various nontrivial mechanical behaviors beyond the reach of existing methods. Scaling up to three-dimensional cellular solids design while considering crystallographic symmetric arrangements could potentially unlock uncommon mechanical functionalities not observed in two dimensional meta-materials. But this requires efficient and faster solvers that our framework suffers from at large number of degrees of freedom in the mechanics modeling. A future work could focus on speeding up the simulations leveraging learned surrogate models (Sun et al., 2021), neural network-based order reduction techniques (Beatson et al., 2020), and also amortized optimization methods (Xue et al., 2020a). Since all crystallographic symmetry groups include translational symmetry, further computational cost reduction can also be achieved by simulating a single unit cell with periodic boundary conditions instead of modeling the entire cellular solid (Mizzi et al., 2021).

ACKNOWLEDGMENTS

We would like to thank Deniz Oktay for his computational mechanics insights. This work was partially supported by NSF grants IIS-2007278 and OAC-2118201, NSF under grant number 2118201, the NSF under grant number 2127309 to the Computing Research Association for the CIFellows 2021 Project. PO is supported by the Gatsby Charitable Foundation.

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

# A APPENDIX

## A.1 THE 17 WALLPAPER GROUPS

The following figure illustrates the 17 distinct (up to isomorphy) space groups on $\mathbb{R}^2$, also known as the wallpaper groups. In each figure, the region marked red corresponds to the polytope $\Pi$ in (1). The letter F is inscribed to mark orientation. Figures from Adams & Orbanz (2023).

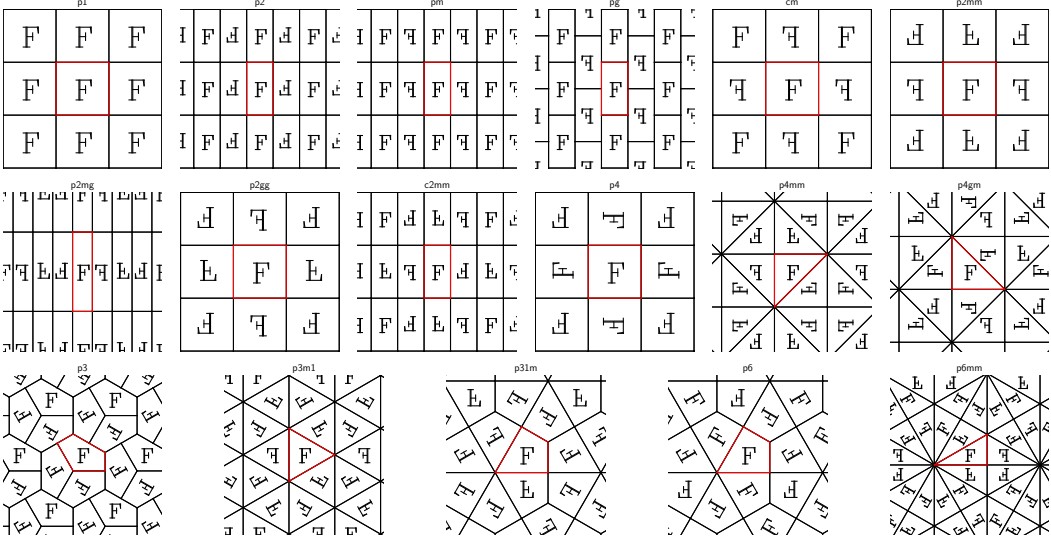

## A.2 PROOFS

### A.2.1 PROOF OF THEOREM 1

This proof draws on one of the fundamental properties of space groups, namely that, if $\mathbb{G}$ is a space group and $\mathbb{T}_{\mathbb{G}}$ its subgroup of pure translations, the quotient group $\mathbb{G}/\mathbb{T}_{\mathbb{G}}$ is finite. See for example (Vinberg & Shvartsman, 1993).

*Step 1.* We first show $\widehat{\mathbb{G}}$ is finite. Consider the relation $\equiv$ on $\mathbb{G}$ defined as

$$\phi \equiv \psi \qquad :\Leftrightarrow \qquad \phi = \psi + \tau \quad \text{for some } \tau \in \mathbb{T}_{\mathbb{G}} \, .$$

Since $\mathbb{T}_{\mathbb{G}}$ is an equivalence relation, and by definition of quotients, the equivalence classes of $\equiv$ correspond to the elements of the quotient group $\mathbb{G}/\mathbb{T}_{\mathbb{G}}$. Since $\mathbb{G}$ is a space groups, this implies $\equiv$ has a finite number of equivalence classes. Recall that each element $\phi$ of $\mathbb{G}$ is of the form

$$\phi(x) \;=\; A_\phi x + b_\phi \;=\; A_\phi x + \sum\nolimits_{i \leq n} c_i(\phi) b_i \, ,$$

where $c(\phi) = (c_1(\phi), \ldots, c_n(\phi))$ is a vector in $\mathbb{R}^n$. It follows that there are unique vectors $\hat{c}(\phi) \in [0,1)^n$ and $\tilde{c}(\phi) \in \mathbb{Z}^n$ such that $c(\phi) = \hat{c}(\phi) + \tilde{c}(\phi)$. The transformations

$$\hat{\phi}(x) \;:=\; A_\phi x + \sum\nolimits_{i \leq n} \hat{c}_i(\phi) b_i \quad \text{and} \quad \tau_\phi(x) \;:=\; x + \sum\nolimits_{i \leq n} \tilde{c}_i(\phi) b_i$$

are hence indeed elements $\hat{\phi} \in \widehat{\mathbb{G}}$ and $\tau_\phi \in \mathbb{T}_{\mathbb{G}}$ that satisfy $\phi = \hat{\phi} + \tau_\phi$, and are the only such elements. It follows that each equivalence class of $\equiv$ contains exactly one element of $\widehat{\mathbb{G}}$, which shows the set $\widehat{\mathbb{G}}$ is indeed finite.

*Step 2.* To verify the properties of $\Gamma$, we must first establish two simple properties of $\widehat{\mathbb{G}}$, namely

$$\text{(i)} \;\; \widehat{\mathbb{G}\psi} = \widehat{\mathbb{G}} \qquad \text{and} \qquad \text{(ii)} \;\; \hat{\phi}\psi(\mathbf{x}) \;=\; \widehat{\phi\psi}(\mathbf{x}) + b_\tau \quad \text{for some } \tau \in \mathbb{T}_0 \, . \tag{9}$$

Here, $\mathbb{G}\psi$ is the set obtained by composing each element of $\mathbb{G}$ with $\psi$ from the right, and $\widehat{\mathbb{G}\psi}$ denotes application of the definition of $\widehat{\mathbb{G}}$ to $\mathbb{G}\psi$. Since $\mathbb{G}$ is a group and $\psi$ one of its elements, we have

$\mathbb{G}\psi = \mathbb{G}$, so property (i) holds. To verify (ii), observe that the "hat operation" $\hat{\phi}$ preserves the equivalence class of $\phi$: Since $\phi = \hat{\phi} + \tau_\phi$ and $\tau_\phi \in \mathbb{T}_\mathbb{G}$, we have $\hat{\phi} \equiv \phi$. If $\psi$ is a further group element, we also have $\hat{\phi}\psi(x) = \phi(\psi x) - b_{\tau_\phi}$ and therefore $\hat{\phi}\psi \equiv \phi\psi$. In summary, we have

$$\hat{\phi}\psi \ \equiv \ \phi\psi \ \equiv \ \widehat{\phi\psi}$$

which shows (ii)

*Step 3.* Now consider a $\mathbb{T}_\mathbb{G}$-invariant function $f$. For any $\psi \in \mathbb{G}$, we have

$$
\begin{aligned}
\sum_{\hat{\phi} \in \widehat{\mathbb{G}}} A_\phi^{-1} f(\hat{\phi}\psi\mathbf{x}) \ &= \ \sum_{\pi \in \widehat{\mathbb{G}}\psi} A_{\pi\psi^{-1}}^{-1} f(\pi\mathbf{x}) && \text{substitute } \pi := \hat{\phi}\psi \\
&= \ \sum_{\pi \in \widehat{\mathbb{G}}\psi} A_{\pi\psi^{-1}}^{-1} f(\hat{\pi}\mathbf{x} + b_\tau) && \text{by (9i), for some } \tau \in \mathbb{T}_0 \\
&= \ \sum_{\pi \in \widehat{\mathbb{G}}\psi} A_{\pi\psi^{-1}}^{-1} f(\hat{\pi}\mathbf{x}) && \text{since } h \text{ is } \mathbb{T}_0\text{-invariant} \\
&= \ A_\psi \sum_{\pi \in \widehat{\mathbb{G}}\psi} A_\pi^{-1} f(\hat{\pi}\mathbf{x}) && \text{since } A_{\pi\psi^{-1}}^{-1} = A_{\psi^{-1}}^{-1} A_\pi^{-1} = A_\psi A_\pi^{-1} \\
&= \ A_\psi \sum_{\hat{\pi} \in \widehat{\mathbb{G}}} A_\pi^{-1} f(\hat{\pi}\mathbf{x}) && \text{by (9ii) .}
\end{aligned}
$$

It follows that

$$(\Gamma f)(\psi x) \ = \ \sum_{\hat{\phi} \in \widehat{\mathbb{G}}} A_\phi^{-1} f(\hat{\phi}\psi\mathbf{x}) \ = \ A_\psi \sum_{\hat{\phi} \in \widehat{\mathbb{G}}} A_\phi^{-1} f(\hat{\phi}\mathbf{x}) \ = \ A_\psi (\Gamma f)(x)$$

for any $\psi \in \mathbb{G}$. $\qquad\square$

### A.2.2 Proof of Theorem 2

The existence and uniqueness of solutions of (4) is given by the Picard-Lindelöf theorem (e.g. Deuflhard & Bornemann, 2002). Here is a statement that suffices for our purposes:

**Fact 1** (Version of the Picard-Lindelöf theorem). *Fix $n, k \in \mathbb{N}$ and a bounded interval $I := [0, t_{max}]$. Let $H : \mathbb{R}^n \times I \to \mathbb{R}^d$ be a Lipschitz function that is $k \geq 1$ times continuously differentiable in its second argument. Then*

$$F(x_0, t) \ := \ x_0 \ + \ \int_0^t H(F(x_0, s), s) ds$$

*defines a function $F : \mathbb{R}^n \times I \to \mathbb{R}^d$ that is continuous in its first argument, $k + 1$ times continuously differentiable in the second, and is the unique solution of* (4).

We break the main ingredients of the proof of Theorem 2 up into three lemmas. First, we show that a function $H$ that satisfies (5) yields flows with the symmetry properties we require.

**Lemma 1.** *Let $\mathbb{G}$ be a group of isometries of $\mathbb{R}^n$, and fix some $t_{\max} > 0$. Let $H : \mathbb{R}^n \times I \to \mathbb{R}^n$ be a function is Lipschitz, and $k$ times continuously differentiable in its second argument, for $k \geq 1$. Then there is a unique function $F : \mathbb{R}^n \times [0, t_{\max}] \to \mathbb{R}^n$ that satisfies (4) for all $(x, t)$ in its domain, and this function satisfies*

$$F(\phi x, t) \ = \ \phi F(x, t) \qquad \text{for all } \phi \in \mathbb{G}, \text{ all } x \in \mathbb{R}^n \text{ and } 0 \leq t \leq t_{\max} \tag{10}$$

*if and only if $H$ satisfies* (5). *The solution $F$ is continuous in its first argument, and $(k + 1)$ times continuously differentiable in the second.*

*Proof.* We first construct the solution $F : \mathbb{R}^n \times I \to \mathbb{R}^n$ on the larger interval $I = [-1, t_{\max}]$. (This is to ensure that $0$ is in the interior of $I$, which we need for step 3 below.) The restriction of $F$ to $\mathbb{R}^n \times [0, t_{\max}]$ is then solution in the statement of the result.

*Step 1.* Existence, uniqueness, and smoothness of $F$ on $\mathbb{R}^n \times I$ then follow immediately from Fact 1, where we set $t_{\min} = -1$ and $t_0 = 0$. Fact 1 also shows that $F$ solves

$$\frac{d}{dt} F(x, t) \ = \ H(F(x, t), t) \quad \text{with} \quad F(x, 0) \ = \ x . \tag{11}$$

What we have to show is that $F$ is equivariant if and only if $H$ satisfies (5).

*Step 2.* Suppose $H$ satisfies (5). Write $F^\phi(x,t) := \phi^{-1}F(\phi x, t)$. Thus, $F$ is equivariant iff $F^\phi = F$ for all $\phi$.

$$
\begin{aligned}
\frac{d}{dt}F^\phi(x,t) &= \frac{d}{dt}(A_{\phi^{-1}}F(\phi x, t) + b_{\phi^{-1}}) &&\text{since } \phi^{-1}x = A_{\phi^{-1}}x + b_{\phi^{-1}} \\
&= A_{\phi^{-1}}\frac{d}{dt}F(\phi x, t) &&A_{\phi^{-1}} \text{ and } b_{\phi^{-1}} \text{ do not depend on } t \\
&= A_{\phi^{-1}}H(F(\phi x, t), t) &&\text{by (11)} \\
&= H(\phi^{-1}F(\phi x, t), t) &&\text{by (5)} \\
&= H(F^\phi(x,t), t) &&\text{definition of } F^\phi.
\end{aligned}
$$

Substituting the initial condition into the definition of $F^\phi$ shows

$$
F^\phi(x,0) = \phi^{-1}(F(\phi x, 0)) = \phi^{-1}\phi x = x
$$

Thus, both $F$ and $F^\phi$ solve the differential equation (11) for the initial condition $x$. Since the solution is unique by Fact 1, it follows that $F(x,t) = F^\phi(x,t)$ for all $t$. In summary, (5) implies equivariance of $F$.

*Step 3.* Conversely, assume $F$ is equivariant, so $F^\phi = F$. Since $H$ is continuously differentiable, Fact 1 shows $t \mapsto F(x,t)$ is twice continuously differentiable. Its Taylor expansion around any $t$ in the interior $(-1, t_{\max})$ is hence

$$
\begin{aligned}
F(x, t+\epsilon) &= F(x,t) + \frac{d}{dt}F(x,t) \cdot \epsilon + o(\epsilon^2) \\
&= F(x,t) + H(F(x,t), t) \cdot \epsilon + o(\epsilon^2) &&\text{for each } x \in \mathbb{R}^n \text{ and } \epsilon > 0 \,.
\end{aligned}
\tag{12}
$$

If we set $t = 0$ and use the initial condition $F(x,0) = x$, we obtain

$$
\phi F(x,\epsilon) = A_\phi(x + H(x,0)\cdot\epsilon + o(\epsilon^2)) + b_\phi = \phi x + A_\phi H(x,0)\cdot\epsilon + o(\epsilon^2)\,,
$$

where we note that $A_\phi o(\epsilon^2) = o(\epsilon^2)$ since $A_\phi$ is an isometry. Substituting $\phi x$ into (12) shows

$$
F(\phi x, \epsilon) = \phi x + H(\phi x, 0) + o(\epsilon^2)
$$

Since $F^\phi = F$, we hence have

$$
\phi x + H(\phi x, 0)\cdot\epsilon + o(\epsilon^2) = \phi x + A_\phi H(x,0)\cdot\epsilon + o(\epsilon^2)
$$

and therefore

$$
H(\phi x, 0) = A_\phi H(x,0) + o(\epsilon)
$$

Since that is true for all $\epsilon$, it follows that $H(\phi x, 0) = A_\phi H(x,0)$. In summary, equivariance of $F$ implies that (5) holds at $t = 0$. Since $F$ is a flow, the flow $F^t(x,s) := F(x, t+s)$ satisfies (4) for the function $H^t(x,s) := H(x, t+s)$ and the initial value $x_0^t := F(x,t)$. If $F$ is equivariant, so is $F^t$. We can hence apply the same argument at $s = 0$, which shows $H(\phi x, t) = A_\phi H(x,t)$. $\qquad\square$

Given a differentiable function $f : \mathbb{R}^n \to \mathbb{R}^m$, we denote by $Df$ its differential. If $m = n$, this is the Jacobian matrix, and if $m = 1$, the transpose $(Df)^\mathsf{T}$ is the gradient of $f$.

**Lemma 2.** *For any differentiable function $g : \mathbb{R}^{2n} \to \mathbb{R}^m$ and any $m \in \mathbb{N}$, the differential $D(g \circ \rho)$ is $\mathbb{T}_\mathbb{G}$-invariant. In particular, $D\rho$ is $\mathbb{T}_\mathbb{G}$-invariant.*

*Proof.* Since the Jacobian matrix of $\rho$ has entries

$$
(D\rho(x))_{ij} = \begin{cases} 2\pi\rho_{2i-1}(x) & \text{if } j = 2i - 1 \\ -2\pi\rho_{2i}(x) & \text{if } j = 2i \\ 0 & \text{otherwise} \end{cases}
$$

it can be written as a function $D\rho = M \circ \rho$ of $\rho$, where $M : \mathbb{R}^{2n} \to \mathbb{R}^{n \times 2n}$. That shows

$$
D(g \circ \rho)(x) = (Dg)(\rho(x)) \cdot (D\rho)(x) = (Dg \cdot M)(\rho(x))\,,
$$

and hence $D(g \circ \rho) \circ \phi = (Dg \cdot M) \circ \rho \circ \phi = (Dg \cdot M) \circ \phi$. $\qquad\square$

Our final auxiliary result shows that the symmetrization operator $\Gamma$ does not increase divergence. For Theorem 2, this means that if a function $f$ is divergence-free, then so is $\Gamma f$.

**Lemma 3** ($\Gamma$ does not increase divergence). *If $\mathbb{G}$ is a space group on $\mathbb{R}^n$, and $f : \mathbb{R}^n \to \mathbb{R}^n$ is a continuously differentiable function, then*

$$\big\| \operatorname{div}(\Gamma f) \big\|_{\sup} \ \leq \ \big\| \operatorname{div} f \big\|_{\sup} .$$

*Proof.* Since the divergence is the trace of the differential, and both trace and differential are linear,

$$\operatorname{div}(\Gamma f)(x) \ = \ \operatorname{tr} D\Big( \tfrac{1}{|\hat{\mathbb{G}}|} \textstyle\sum_{\phi \in \hat{\mathbb{G}}} A_\phi^{-1} f(\phi x) \Big) \ = \ \tfrac{1}{|\hat{\mathbb{G}}|} \textstyle\sum_{\phi \in \hat{\mathbb{G}}} \operatorname{tr} D\big( A_\phi^{-1} f(\phi x) \big)$$

Each summand has differential

$$D(A_\phi^{-1} f \circ \phi)(x) \ = \ A_\phi^{-1}(Df)(\phi x)(D\phi)(x) \ = \ A_\phi^{-1}(Df)(\phi x) A_\phi$$

Since the trace is linear and orthogonally invariant, it follows that

$$|\operatorname{tr} D(A_\phi^{-1} f \circ \phi)(x)| \ = \ |\operatorname{tr}((Df)(\phi x))| \ = \ |\operatorname{div}(f)(\phi x)| \ \leq \ \|\operatorname{div}(f)\|_{\sup}$$

The result then follows via the triangle inequality,

$$\|\operatorname{div}(\Gamma f)(x)\|_{\sup} \ = \ \tfrac{1}{|\hat{\mathbb{G}}|} \textstyle\sum_{\phi \in \hat{\mathbb{G}}} \|\operatorname{div} f\|_{\sup} \ = \ \|\operatorname{div} f\|_{\sup} ,$$

which is all we had to show. $\qquad\square$

*Proof of Theorem 2.* Since $\rho$ is $\mathbb{T}_{\mathbb{G}}$-invariant and infinitely often differentiable, the function $h \circ \rho$ is $\mathbb{T}_{\mathbb{G}}$-invariant and $(k+1)$ times continuously differentiable. The function $\Lambda(h \circ \rho)$, by definition of $\Lambda$, a function of the gradient $\nabla(h \circ \rho)$. Since this gradient is $\mathbb{T}_{\mathbb{G}}$-invariant by Lemma 2, $\Lambda(h \circ \rho)$ is $\mathbb{T}_{\mathbb{G}}$-invariant. By Theorem 1, $\Gamma\Lambda(h \circ \rho)$ satisfies (5). It is also divergence-free, since $\Lambda(h \circ \rho)$ is divergence-free and $\Gamma$ does not increase divergence, by Lemma 3. We can therefore choose $H := \Gamma\Lambda(h \circ \rho)$ in Lemma 1, and the lemma shows that there is a unique solution $F$ that is an equivariant flow. Since $H$ is also divergence-free, as suitable version of Liouville's theorem (e.g. Arnold (1998), Theorem B.3.1) shows $F$ is volume-preserving. $\qquad\square$

### A.3 MECHANICAL MODEL

We use a nearly incompressible neo-Hookean hyperelastic model, which effectively describes the mechanical behavior of the elastomeric materials commonly employed in manufacturing of flexible meta-materials. The mechanical behavior of such hyperelastic material can be described by introducing a strain energy density function $\boldsymbol{\Psi}$ that is independent of the path of deformation and is a function of the deformation gradient tensor $\mathbf{F} = \nabla\mathbf{u} + \mathbf{I}$. Here, $\mathbf{u}$ is the displacement field, which is a function $\mathbf{u} : \Omega \to \mathbb{R}^2$. A representation of this function is computed by the differentiable mechanics solver described in Section 4. Since the mesh representation of the shape $s_\theta$ uses $N$ mesh points, and the vector field has two degrees of freedom at each point, the representation of $\mathbf{u}$ is an element of $\mathbb{R}^{2N}$. With these ingredients, the function $\boldsymbol{\Psi}$ is given by

$$\boldsymbol{\Psi}(\mathbf{F}) = \frac{\mu}{2}\left( \operatorname{tr}(\mathbf{F}^T\mathbf{F}) - 2 - 2\ln\det(\mathbf{F}) \right) + \frac{\lambda}{2}\left( \ln\det(\mathbf{F}) \right)^2, \tag{13}$$

where $\mu = E/2(1+\nu)$ and $\lambda = \nu E/(1+\nu)(1-2\nu)$ are Lamé parameters of a material with Young's modulus $E$, and $\nu$ is the Poisson's ratio. The displacement field of the structure at the equilibrium $\mathbf{u}^*$ is computed from the space of all admissible displacement fields $\mathcal{H}$ that satisfy Dirichlet boundary conditions (that is, $\mathbf{u}$ takes a suitable prescribed value on the boundary), by minimizing the total stored potential energy from material deformation under mechanical loading

$$\mathbf{u}^* = \arg\min_{\mathbf{u}\in\mathcal{H}} \int_\Omega \boldsymbol{\Psi}(\mathbf{F}) dX. \tag{14}$$

The minimizer is computed using a second order Newton method for sparse Hessians.

### A.4 MESH INDEPENDENCE ANALYSIS

We performed mesh independence analysis for all the cases to ensure simulation results from mechanics solver were not affected by insufficient mesh resolution. The details of an example of these analyses with associated time for each optimization step is shown in Table 2 for designing under uniaxial tension of `p1` group. Mesh independence is studied in an initial reference model.

Table 2: Mesh independence analysis based on the deformation of the middle right edge of a cellular solid with `p1` group symmetry under uniaxial tension.

| number of DoF | 9,000 | 14,000 | 20,000 | 25,000 | 29,000 |
|---|---|---|---|---|---|
| middle deformation (mm) | 3.05 | 3.31 | 3.49 | 3.65 | 3.65 |
| optimization time (s) | 15 | 34 | 49 | 61 | 95 |

### A.5 META-MATERIAL DESIGNS WITH P1 SYMMETRY UNDER UNIAXIAL TENSION

Three cellular solids designs while pores respecting `p1` symmetry group under uniaxial tension when aiming for linear responses with $\beta$ equals to $0.1$, $0.5$, and $1.5$. The design 4 is with `p1` symmetry group when targeting linear response with $\beta = 1.5$.

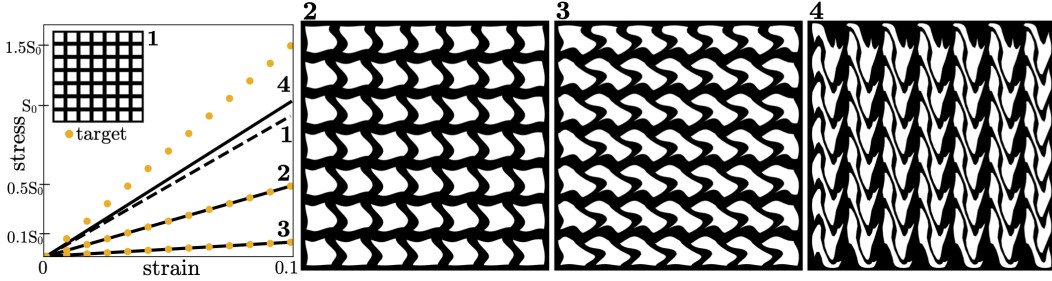

### A.6 META-MATERIAL DESIGNS WITH NEGATIVE POISSON'S RATIOS

All eight cellular solids designs with best performances in achieving $\nu_{ef} = -0.5$ under uniaxial tension while pore shapes respecting different symmetry groups.

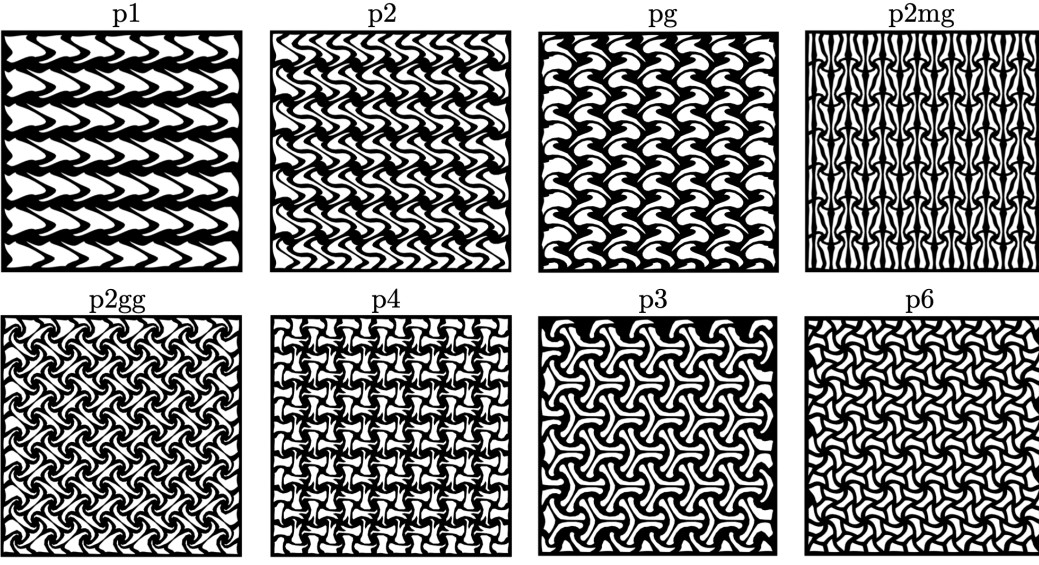

### A.7 Fabrication and Experimental Details

140mm-by-140mm samples with pull tabs were laser cut from butyl-based laser-engravable rubber sheets of 2.38 mm thickness. Laser cutting was performed using a Universal Laser Systems PLS6.150D machine with two 75-W $CO_2$ lasers emitting at 10.6 $\mu$m. Cut patterns were programmed using the Universal Laser Systems software suite with the following settings: 3 % travel speed, 20 % power. Cutting was repeated three times.

Experimental tensile tests were performed using an Instron 5969 UTM outfitted with an Instron 2530-series 50N static load cell. Samples were mounted in the UTM's clamps by their pull tabs; a controlled displacement of the top clamp between $-5$mm and 20mm was applied over the course of three cycles. Video footage of the sample was used to extract the vertical and lateral strain of each sample midway between the clamps, allowing calculation of the average experimental Poisson's ratio $\nu_{ex}$ at 0.1 vertical strain.

### A.8 Meta-Material Designs under Uniaxial Compression

Examples of cellular solids designs with target force-displacement responses under uniaxial compression respecting symmetry groups `pm`, `p2mg`, and `p4` that do not match target values. These simulations ran with several different initialization of the neural network parameters.

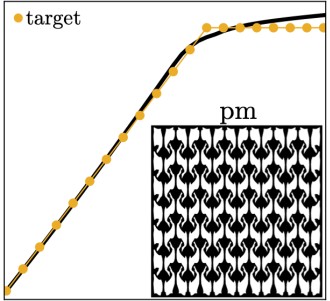 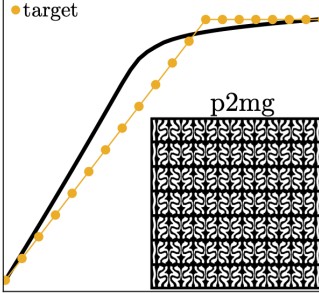 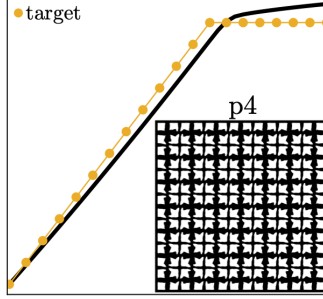

