# OpenReview forum: "Designing Mechanical Meta-Materials by Learning Equivariant Flows"
_ICLR.cc/2025/Conference — ICLR 2025 Poster_

### Official Review · Reviewer_4yCV · 2024-10-22

**Soundness:** 2
**Presentation:** 2
**Contribution:** 2
**Rating:** 5
**Confidence:** 3

**Summary:**

The authors present a machine learning framework that efficiently tackles the inverse design of two-dimensional cellular solids. By training a neural network to learn a symmetry-preserving flow, they could expand the design space of a class of mechanical metamaterials, known as cellular solids, into more generalized isometries. Their approach ensures that the solution avoids disconnected regions and the associated numerical challenges. Moreover, they demonstrate that such neural flows are divergence-free, enabling the preservation of the total volume while focusing solely on optimizing the geometry. The effectiveness of their approach is validated  through simulations and the fabrication of real-world prototypes.

**Strengths:**

- This paper demonstrates how to expand the design space of cellular solids beyond translational symmetry using symmetry-preserving flow. It also provides detailed derivations and proofs.
- This paper leverages a neural network to construct the symmetry-preserving flow and optimizes materials to achieve desirable mechanical properties. The effectiveness of the approach has been validated through simulations and real-world fabrication.

**Weaknesses:**

- The contributions and related work are not clearly presented. Does this optimization formulation for extending the design space already exist, but cannot be effectively solved by classical optimization methods, or is it first introduced in this paper? Are there any other formulations or approaches that can also extend the design space of cellular solids? If so, why is this formulation considered superior to others?
- My main concern is the motivation of using a neural network for such design problem. Many other machine learning methods could also satisfy the conditions outlined in Theorem 2 and the loss function. Why is the neural network preferred? There is no justification, proof, or experimental comparison to support this choice of neural networks as a proxy model.  This issue downsides this paper quality significantly. I strongly suggest the authors provide strong motivation from this aspect.

**Questions:**

- It appears that the neural network is not always Lipschitz-continuous or ($k+1$)-times continuously differentiable. Is there any proof supporting this, or did you relax the constraint based on certain assumptions?
- It is highly suggested to compare the proposed approach to other machine learning alternatives. These models can also satisfy the conditions outlined in Theorem 2 and offer greater interpretability compared to MLP.
- How did you determine the configuration of the neural network? Does the configuration—such as the number of hidden neurons, layers, or the choice of activation function—affect the simulation accuracy? Additionally, which components of the neural network play the most critical role in the simulation performance? Whether the learned function exactly meets the constraint stated in Theorem 2?

---

> ### Author Response · Authors · 2024-11-24
>
> Thank you very much for reviewing our paper!
>
> $\textbf{Is the method first introduced in this paper?}$
>
> Our proposed method, to the best of our knowledge, is first in its kind and none of the existing approaches investigate systematically the effect of pores arrangement on meta-materials mechanical responses. While current methods focused on translational symmetries of a single pore unit cell to construct cellular solids, we developed a machine learning framework that enables a systematic approach to design cellular solids with all 17 2D crystalograhic arrangement of the pores. We showed several designs beyond the reach of translational symmetry. We also modified the introduction and related works sections for more clarity.
>
> $\textbf{Neural network architecture and continuity}$
>
> We tested more complex neural network architectures with more hidden layers, more neurons, and different activation functions. But we observed nontrivial changes in the final designs when increasing number of hidden layers and neurons. We also found that $\tanh$ nonlinearity leads to a faster convergence of the optimization process among other activation functions we tested e.g., sine function. The simplest neural network architecture that consistently proposes meta-material designs in all 17 symmetry groups is reported in Line 319 of the original paper (fully-connected neural network with two hidden layers of size 10, with $\tanh$ nonlinearity).
>
> We like to point that the proposed network is continuous and able to propose all designs cases we aimed for. However, there could be some other machine learning alternatives that can also work efficiently in our computational framework.

---

> > ### Comment · Reviewer_4yCV · 2024-11-27
> > **Response to rebuttal**
> >
> > Thank the authors for some clarification.
> >
> > However, I do not think the rebuttal can address my concerns, particularly on how the traditional numerical approaches solve this problem and how your method compares with them in term of accuracy and efficiency. I think this point is the basic requirement for a machine learning-based approach to solve a science problem. Then what about other machine learning approaches besides neural networks.
> >
> > I think the paper lacks some criticality regarding model comprison. I tend to keep the score.

---

> ### Comment · Area_Chair_fhjF · 2024-11-27
> **Response**
>
> Dear Reviewer,
> Do you mind letting the authors know if their rebuttal has addressed your concerns and questions? Thanks!
> -AC

---

> ### Author Response · Authors · 2024-11-27
>
> Thanks for your comments. We added some additional clarifications.
>
> We modified the introduction and the related works sections to improve the clarity. But we briefly discuss them here: Cellular solids are porous structures with the property that the geometric features of the pores define their mechanical responses, rather than their chemical composition. Therefore, the nonlinear functionalities of cellular solids can be programmed by: i) optimizing the geometry of the pores, and/or ii) manipulating the ``arrangement" of the pores. Existing design approaches have been merely focused on the former, optimizing the pore shape of a unit cell that is used to construct a cellular solid with its translational symmetry. However, manipulating of the arrangement (that has not been studied before) can lead to uncommon mechanical behaviors (as we showed in this paper). In addition, the existing approaches use level-set method or moving morphable voids method which suffer from designs with disconnected regions and the associated numerical challenges. Our contribution is: we proposed a novel machine learning framework, the first of its kind, for learning richer classes of cellular solids, in which we not only optimize the pore shapes but also explore all possible arrangements among two-dimensional crystallographic symmetry groups. Moreover, another advantage of our proposed approach is that it ensures the solution avoids disconnected regions and associated numerical challenges via leveraging an equivariant and divergence-free neural network flow. We showed several designs from 17 2D symmetry groups that can significantly outperform designs with p1 symmetry (translational symmetry). For example, linear force-displacement responses under tension and Negative Poisson's ratio under tension and compression.
>
> Please find the revised version of the Introduction section for more details.
>
> $\textbf{No comparison with existing approaches}$
>
> Our proposed method is the first of its kind, to the best of our knowledge, that enables a systematic approach to study the effect of pores ``arrangement" on mechanical responses of cellular solids. The most effective way for comparison we realized is to compare results from all wallpaper symmetry groups with cellular solids constructed with translational symmetry (p1 symmetry). This comparison is provided in all the results sections. We showed that designs with other 16 symmetry groups outperform the design with translational arrangement except in one case. This observation confirms that modifying both pore shapes and pore arrangement simultaneously enable a drastic expansion of the design space of cellular solids.

---

### Official Review · Reviewer_DtCf · 2024-11-03

**Soundness:** 3
**Presentation:** 3
**Contribution:** 3
**Rating:** 8
**Confidence:** 3

**Summary:**

The paper presents a method to design metamaterials by learning equivariant flows. The method is capable of designing materials for all 17 space groups in two dimensions.

**Strengths:**

The paper is well written and the presented approach is mathematically grounded. The problem of designing mechanical metamaterials is an active area of research. The work presented here involves metamaterials where the unit cell is a general porous structure where each point in the domain can be air $(\rho=0)$ or material $(\rho=1)$. This is a good focus at times when much of the work in the field is on strut-based lattice metamaterials.

The idea of starting out with an initial model with well-behaved connected topology and morphing it into a flowed model is important – for a well-behaved flow, one will not end up with defective topology and disconnected patches of material (as opposed to what one might obtain by using fully generative models such as diffusion).

Part of having a well-behaved flow is ensuring it is volume-preserving. The authors incorporated an idea from the literature in which a divergence-free flow is obtained as vector whose components are appropriately arranged partial derivatives of a scalar function. This enables to set an original relative density $\bar{\rho}$ which is then preserved by the flow.

**Weaknesses:**

1. The presented approach is in 2d which has limited relevance as we often want materials with 3d structure (not extrusions of 2d cross-sections)

2. The emphasis on validation with real-world prototypes is only tangentially relevant to the presented work.
E.g. in abstract: “We validate these new designs in simulation and by fabricating real-world prototypes.” There are two statements to unpack:
    1.	validate these new designs in simulation - this is not really validation since the training loop already involves the differentiable simulator
    2. validate these new designs by fabricating real-world prototypes – this is not really the validation of the presented equivariant flows, but rather a validation that the differentiable mechanics simulator captures real-world behaviour of the structures

    Overall please change the way in which such claims are formulated:
        (i)	any claimed “validation of simulations with respect to target behaviour” is only a test that the target behaviour is within the expressive power of the framework. It would be interesting to push this to the limit – for instance, what would happen if you tried to design for tension while setting as the target the same stress-strain curve which you use in compression?
        (ii)	“validation by manufacturing real-world specimens” is only a check of the ability of the differentiable mechanics simulator to capture real-world behaviour of the structures.

3. One limitation identified in section 7 for scaling up to 3d structures is the lack of efficient/fast solvers and the high number of degrees of freedom. In section 5 it is mentioned that 25000 degrees of freedom were needed for force-displacement experiments. Please include the convergence study with respect to mesh resolution (accuracy/stability vs time per iteration vs number of DoFs)

**Questions:**

The following typos were found:

-	125: grammar of the sentence “In the case…”
-	354: “preserved”
-	531: grammar of the sentence
-	849: grammar: “vector field has to degrees”

Please compress the pdf; the current version has 27.5MB. It can be compressed without noticeable loss of quality to ~3MB.

Please make it clear in Fig.1 (visually or in caption) that the columns of (b) correspond to a single sample while rows are different stages of deformation

Please show in the plots examples of the behaviour for cases when the target was not obtained (Design under uniaxial compression for symmetry groups other than p1, Design under uniaxial tension for p1 and $\beta=1.5$)

---

> ### Author Response · Authors · 2024-11-24
>
> Thank you very much for reviewing our paper!
>
> $\textbf{2D meta-materials}$
>
> We agree that the design space of two-dimensional (2D) cellular solids is inherently limited, whereas three-dimensional (3D) meta-materials have the potential to offer a broader range of functionalities. However, we believe for this paper 2D cellular solids were a great ansatz, where we showed several interesting nontrivial mechanical responses with our 2D designs. We would like to note that our equivariant neural network flow implementation is capable of designing 3D cellular solids. But our continuum mechanics solver have trouble at large numbers of degrees of freedom, due to the iLU factorization we perform to precondition the iterative solver. Future extensions include modern multigrid methods to scale our solver up or speeding up the simulations leveraging learned surrogate models, and neural network-based order reduction techniques.
>
> $\textbf{Validation with real-world prototypes}$
>
> We agree that the way we describe validation with real-world prototypes is confusing especially in the abstract. We are actually validating the simulated mechanical responses of the designed model against the behavior of fabricated meta-material counterparts in real-world experiments. to clarify this, we modified the sentence in the abstract to ``We validate simulated mechanical behaviors of these new designs against fabricated real-world prototypes."
>
> $\textbf{Convergence study}$
>
> Thanks for pointing this out. We included the convergence study with respect to mesh resolution in the appendix (Appendix 4) and we modified line 359 of the original version of the paper to ``there are at least $\sim$25,000 degrees of freedom in all mechanics models that found satisfactory through a mesh independence analysis described in Appendix A.4, such that simulation results were not affected by insufficient discretization resolution."

---

> ### Author Response · Authors · 2024-11-24
>
> $\textbf{Typos}$
>
> All the typo suggestions are fixed now.
>
> $\textbf{Please compress the pdf}$
>
> Although we added new figures to the paper, we reduced the size to $\sim$12 MB while maintaining the quality.
>
> $\textbf{Please make it clear in Fig.1 (visually or in caption) that the columns of (b) correspond to a single sample while rows are different stages of deformation.}$
>
> We clarified this now visually in Fig. 1.
>
> $\textbf{Please show in the plots examples of the behaviour for cases when the target was not obtained (Design under uniaxial compression for symmetry groups other than p1, Design under uniaxial tension for p1 and $\beta=1.5$}$
>
> We included a new section in the appendix (Appendix 8) with figures depicting designs other than p1 symmetry group that were not able to produce target force-displacements curve under uniaxial compression. And also designs for p1 with $\beta=1.5$ under uniaxial tension are added in Appendix 5.

---

> ### Comment · Area_Chair_fhjF · 2024-11-27
> **Response**
>
> Dear Reviewer,
> Do you mind letting the authors know if their rebuttal has addressed your concerns and questions? Thanks!
> -AC

---

> ### Comment · Reviewer_DtCf · 2024-11-27
>
> I thank the authors for addressing my comments. They were sufficiently addressed.
> I think that this is a good paper with valuable contributions and I have thus raised my score.

---

### Official Review · Reviewer_Z228 · 2024-11-04

**Soundness:** 3
**Presentation:** 4
**Contribution:** 3
**Rating:** 6
**Confidence:** 2

**Summary:**

This paper explores using symmetry preserving flows and a mechanical simulator in order to design new meta-materials with desirable properties. The authors demonstrate how to construct neural networks equivariant to space groups and show successful designs that exhibit good force-displacement responses and effective Poisson ratios. They also manufacture real world designs and validate the simulation results on them.

**Strengths:**

- Novel application of equivariant/invariant networks to meta-materials design
- The paper is clearly written and explains various meta-materials related terms for the unfamiliar reader.
- The authors describe, in great detail, the space groups and the construction of the symmetry preserving flows
- The experiments seem to show good results and the authors further experiment in the real world

**Weaknesses:**

I imagine that there could be multiple designs that satisfy the desired properties. However the proposed method doesn't seem to allow for generating multiple different valid designs as it is not a generative model. The space group and the desired property must first be fixed, and as the network trains, it converges to some design that optimizes the loss. There doesn't seem to be an easy way to generate multiple valid designs without training every time.

One other concern is that I wonder if this paper is suitable for this venue. As a reviewer who doesn't know much about meta-materials design, it is difficult for me to judge the impact of the proposed method and the results. The best I can tell about the results is that the manufactured design seems to match the simulation measurements.

**Questions:**

- Line 319: Is there any specific reason why such small networks were used?

---

> ### Author Response · Authors · 2024-11-24
>
> Thank you for your review of our paper.
>
> $\textbf{Not a generative model}$
>
> We agree that there could be multiple designs that satisfy the desired properties. Our proposed method is not a generative model and we would like to elaborate on this. Generative modeling has been used to tackle challenging design problems in various engineering fields such as designing molecular structures, proteins, and mechanical CAD models. However, generative ML models may struggle to enforce physical, functional, or manufacturability constraints intrinsic to mechanical systems we are interested in our paper. When designs are well-represented as a parametric family (such as implicitly defining the design domain in this work), direct optimization is often a highly-effective way to perform rational design. This approach has a long history in computational mechanics via, e.g., the adjoint method, but advances in machine learning have dramatically increased its potential. Modern automatic differentiation tools such as JAX make it possible not only to construct differentiable physics solvers, but they also enable close coupling with ML frameworks, which can significantly enhance the design capability as shown in this paper. We note that to make sure multiple designs (if needed), we explore among all wallpaper symmetry groups.
>
> $\textbf{Suitable for this venue?}$
>
> In the call for papers, ICLR accepts ``applications to physical sciences (physics, chemistry, biology, etc.)". We believe this is a solid application paper with a novel setup that has demonstrated good, interesting, and unintuitive results to engineering designs. We believe we have introduced a new machine learning framework that leverages equivariant flow to design cellular meta-materials. We tried to have detailed clarification about meta-materials related terms for the unfamiliar reader as you highlighted as the strength of the paper. With regard to the impact to meta-material design community, we showed that our proposed framework drastically expand the design space of cellular solids. While existing approaches merely focused on optimizing the shape/topology of the pores, our proposed method also enables (a systematic approach of) exploring various arrangements of the pores structure. As confirmed by our results, this approach enables designs beyond the reach of existing methods. Just to clarify, the existing approach is focused on designing cellular solids constructed via translational symmetric arrangement (p1 group) of a single unit cell. However, we showed several designs with other symmetry groups that can significantly outperform designs with p1 group. For example, linear force-displacement responses under tension and Negative Poisson's ratio under tension and compression. In addition, our framework ensures the solution avoids disconnected regions and associated numerical challenges observed in existing approaches such as level-set method via implicitly defining the geometry by a neural network flow.

---

> ### Author Response · Authors · 2024-11-24
>
> $\textbf{Is there any specific reason why such small networks were used?}$
>
> We tested more complex neural network architectures with more hidden layers, more neurons, and different activation functions. But we observed nontrivial changes in the final designs when increasing number of hidden layers and neurons. We also found that $\tanh$ nonlinearity leads to a faster convergence of the optimization process among other activation functions we tested e.g., sine function. The simplest neural network architecture that consistently proposes meta-material designs in all 17 symmetry groups is reported in Line 319 of the original paper (fully-connected neural network with two hidden layers of size 10, with $\tanh$ nonlinearity).

---

> ### Comment · Area_Chair_fhjF · 2024-11-27
> **Response**
>
> Dear Reviewer,
> Do you mind letting the authors know if their rebuttal has addressed your concerns and questions? Thanks!
> -AC

---

> ### Comment · Reviewer_Z228 · 2024-11-27
>
> I thank the authors and their explanations have addressed all of my concerns. I have raised my score.

---

### Official Review · Reviewer_rTA7 · 2024-11-10

**Soundness:** 2
**Presentation:** 1
**Contribution:** 2
**Rating:** 5
**Confidence:** 3

**Summary:**

The paper is unclear, but to the best I can understand, the authors investigate an approach to perform inverse design of mechanical metamaterials by utilizing crystallographic symmetry groups.

**Strengths:**

1) Methods to solve inverse problems in the sciences is an important open problem.
2) Building models/methods that leverage symmetry is potentially interesting approach.

**Weaknesses:**

1) Lack of clarity.  Based upon the introduction, I have only a very vague idea of what the authors are proposing, or why they are proposing it.  Specifically, the problem setting is not clear, there is little/no discussion of existing methods to solve the problem, and the limitations of existing methods are also unclear.  These problems make it very difficult to review the work.
a) Line 39: "Existing approaches to cellular solids..".  What is an "approach" to cellular solids referring to?...an approach to what?
b) Line 45: What is "shape optimization"?  We're optimizing the shape for what purpose? This is machine learning conference, and this topic might not be clear to your audience (e.g., me).
c) Line 48: The authors begin talking about their solution, but it is unclear at this point (i) what problem they're trying to solve, (ii) what approaches exist to solve this problem, and (iii) how using "flow" is going to mitigate these problems.
d) Line 48:  What exactly is "flow" in this context?   I see there is a "Related Work" section at the end of the paper, and it cites three papers that the authors claim have studied "Neural network flows" in recent years, however, I looked at the most recent of these cited references, Cranmer et al. 2020, and the word "flow" is never mentioned.  It remains unclear to me what exactly the authors are discussing.

2) The authors present some results with their approach, but there is no mention or comparison with any other existing approaches.

**Questions:**

1) Could the authors explain more clearly what general problem they are trying to solve?  Presumably there is some data-driven learning problem so that this work is suitable for this research community: could the authors explain more clearly what it is?

2) Why don't the authors discuss, or compare with, any other existing approaches for metamaterial design? There are a large number of data-driven inverse design approaches.  Perhaps there is a good reason, but given the problem I cited in (1), it is unclear to me.

See the following reference for some examples, but there are many:  Ardizzone, Lynton, et al. "Analyzing inverse problems with invertible neural networks." arXiv preprint arXiv:1808.04730 (2018).

---

> ### Author Response · Authors · 2024-11-24
>
> Thank you for reviewing our paper.
>
> $\textbf{Lack of clarity}$
>
> We modified the introduction and the related works sections to improve the clarity. But we briefly discuss them here: Cellular solids are porous structures with the property that the geometric features of the pores define their mechanical responses, rather than their chemical composition. Therefore, the nonlinear functionalities of cellular solids can be programmed by: i) optimizing the geometry of the pores, and/or ii) manipulating the ``arrangement" of the pores. Existing design approaches have been merely focused on the former, optimizing the pore shape of a unit cell that is used to construct a cellular solid with its translational symmetry. However, manipulating of the arrangement (that has not been studied before) can lead to uncommon mechanical behaviors (as we showed in this paper). In addition, the existing approaches use level-set method or moving morphable voids method which suffer from designs with disconnected regions and the associated numerical challenges. Our contribution is: we proposed a novel machine learning framework, the first of its kind, for learning richer classes of cellular solids, in which we not only optimize the pore shapes but also explore all possible arrangements among two-dimensional crystallographic symmetry groups. Moreover, another advantage of our proposed approach is that it ensures the solution avoids disconnected regions and associated numerical challenges via leveraging an equivariant and divergence-free neural network flow. We showed several designs from 17 2D symmetry groups that can significantly outperform designs with p1 symmetry (translational symmetry). For example, linear force-displacement responses under tension and Negative Poisson's ratio under tension and compression.
>
> Please find the revised version of the Introduction section for more details.
>
> $\textbf{No comparison with existing approaches}$
>
> Our proposed method is the first of its kind, to the best of our knowledge, that enables a systematic approach to study the effect of pores ``arrangement" on mechanical responses of cellular solids. The most effective way for comparison we realized is to compare results from all wallpaper symmetry groups with cellular solids constructed with translational symmetry (p1 symmetry). This comparison is provided in all the results sections. We showed that designs with other 16 symmetry groups outperform the design with translational arrangement except in one case. This observation confirms that modifying both pore shapes and pore arrangement simultaneously enable a drastic expansion of the design space of cellular solids.
>
> $\textbf{Not a data-driven approach}$
>
> We would also like to note this approach is not a data-driven method, instead we utilized direct optimization technique as a highly-effective way to perform rational design when the design domain is well-represented parametrically (as we implicitly defined design domain via neural network flow).

---

> > ### Comment · Reviewer_rTA7 · 2024-12-01
> > **Thank you for revisions, and my response**
> >
> > I appreciate the authors' addressing my questions.  Although I still find the presentation somewhat unclear, I now think I understand the goals of the work better.  I find the general idea interesting, but because I still find the presentation quite unclear at times (e.g., see example below), and also don't fully understand some portions of the paper, I am uncomfortable giving an "accept" level score.  However, I also don't feel well-positioned to push for a rejection either on the basis of an unclear exposition if other reviewers understand it better and feel it is acceptable.  Therefore I will revise my score to be more neutral.
> >
> > Line 44 - 47: The authors write "recent developments in machine learning of crystallographically-invariant functions (Adams & Orbanz, 2023) make it possible to construct unconstrained parameterizations that can be used for shape optimization. In this work we extend these models... "   The authors don't explain what "these models" are, or their limitations.   Few readers will be familiar with Adams & Orbanz, 2023, which is an arxiv paper from mid-2023 with 3 citations.   If your key contribution is to extend the work in Adams & Orbanz, then you need to explain precisely what they did, its limitations, and precisely how you'll be addressing those limitations.

---

> ### Comment · Area_Chair_fhjF · 2024-11-27
> **Response**
>
> Dear Reviewer,
> Do you mind letting the authors know if their rebuttal has addressed any of your concerns or questions? Thanks!
> -AC

---

### Author Response · Authors · 2024-11-24

We thank all reviewers for their time and feedback on our submission.

To address the reviewer’s comments, we have uploaded a revised version of our manuscript, and we respond to each of the reviewer’s comments individually.

---

### Meta-Review · Area_Chair_fhjF · 2024-12-18

**Metareview:**

**Summary** This paper proposes a method for designing a class of meta-materials called cellular solids.  The method involves an equivariant neural network which defines a volume preserving flow which flows a reference geometry to a new structure for the solid. The structure is optimized to exhibit certain physical properties predicted by a differentiable nonlinear mechanics simulator.

**Strengths**  The paper provides a novel solution to an important problem in meta-materials design and solving inverse problems generally.  The approach is more flexible to previous approaches with a larger design space.  The equivariance constraints allow for the design of materials with different space group symmetries.  The divergence free flow and reference geometry ensure the resulting topology is connected and has the desired volume. Most reviewers felt the paper was clear and well-written after revision. The method is grounded in mathematical theory presented in the paper and validated through simulations and real world tests.

**Weaknesses** A common concern about the paper was the lack of comparisons to alternate methods.  The authors have made the point that their method is the only method which enforces the given constraints with the same design space.  I am somewhat skeptical (along with 4yCV) that there is no way to set up a comparison with alternate design strategies which may or may not use neural networks or to ablate the proposed method to consider the impact of design choices. While lack of clarity was a concern for some reviewers, these concerns were either resolved during revision or not widely shared.  Other concerns focused on limitations with the method: it is only in 2D and it cannot represent the full space of structures which optimize the given problem.  I don’t believe either of these limitations should prevent the publication of this study in its current scope.

**Conclusion** This is an interesting, mathematically-grounded, and novel approach to an important application.  The validation shows it is effective.  I believe the concerns and limitations do not outweigh the contributions.

Thank you to the authors and reviewers for engaging in the review process.

**Additional Comments On Reviewer Discussion:**

Z228 felt their concerns on venue and lack of uniqueness of the solution were addressed and raised their score from 5 to 6.  DtCf raised their score to 8 after the authors responded to their clear and constructive feedback on paper clarity, and the clarity of validations.  4yCV engaged the reviewers in discussion, but was not convinced to increase from 5. They believe the paper does not have adequate comparison to non-NN approaches. DtCf gave the strongest score in favor of the paper and I believe argued convincingly for its acceptance despite concerns on adequate comparison and clarity.

---

### Decision · Program_Chairs · 2025-01-22

Accept (Poster)